# Study on Saccharide–Glucose Receptor Interactions with the Use of Surface Plasmon Resonance

**DOI:** 10.3390/ijms242216079

**Published:** 2023-11-08

**Authors:** Maciej Trzaskowski, Marcin Drozd, Tomasz Ciach

**Affiliations:** 1Centre for Advanced Materials and Technologies CEZAMAT, Warsaw University of Technology, Poleczki 19, 02-822 Warsaw, Poland; marcin.drozd.ch@pw.edu.pl; 2Faculty of Chemistry, Warsaw University of Technology, Noakowskiego 3, 00-664 Warsaw, Poland; 3Faculty of Chemical and Process Engineering, Warsaw University of Technology, Waryńskiego 1, 00-645 Warsaw, Poland; tomasz.ciach@pw.edu.pl

**Keywords:** surface plasmon resonance, biosensors, angiostatics, nanoparticles, saccharides

## Abstract

The aim of this study was to investigate the process of attachment of saccharide particles differing in degree of complexity to cell receptors responsible for transport of glucose across the cell membrane (GLUT proteins). This phenomenon is currently considered when designing modern medicines, e.g., peptide drugs to which glucose residues are attached, enabling drugs to cross the barrier of cell membranes and act inside cells. This study aims to help us understand the process of assimilation of polysaccharide nanoparticles by tumour cells. In this study, the interactions between simple saccharides (glucose and sucrose) and dextran nanoparticles with two species of GLUT proteins (GLUT1 and GLUT4) were measured using the surface plasmon resonance technique. We managed to observe the interactions of glucose and sucrose with both applied proteins. The lowest concentration that resulted in the detection of interaction was 4 mM of glucose on GLUT1. Nanoparticles were measured using the same proteins with a detection limit of 40 mM. These results indicate that polysaccharide nanoparticles interact with GLUT proteins. The measured strengths of interactions differ between proteins; thus, this study can suggest which protein is preferable when considering it as a mean of nanoparticle carrier transport.

## 1. Introduction

The process of glucose transport via cell membranes—one of the most important processes of cell nutrition—is mediated by intermembrane proteins from the GLUT (glucose transporters) family, which consists of 13 known proteins (GLUT 1–12 and HMIT1) differing slightly in structure and specific function. In addition to glucose transport, these proteins probably facilitate the membrane transport of other substances, provided that their particles contain β-D-glucose residues to which GLUT proteins exhibit strong affinity. This phenomenon is currently considered when designing modern medicines, e.g., peptide drugs to which glucose residues are attached, thus enabling drugs to cross the barrier of cell membranes and act inside cells. The mechanism of this interesting phenomenon has not yet been accurately described. The main question is the size limitation of a particle that can be transported this way and how does the size of the attached saccharide affect the efficiency of the process of binding molecules with GLUT proteins [1]. Nanoparticles are defined as colloidal particles with diameters below 1000 nm. Currently, they are obtained from metals (e.g., gold, silver), natural and synthetic polymers and are used for both diagnostic and therapeutic purposes. When intended for use as carriers of anticancer drugs, the particle diameter should not exceed 100 nm to minimise the organism’s immune response, increase blood circulation time, and maximise passive diffusion through the porous structure of the blood vessels formed during angiogenesis. However, a crucial factor determining the fate of a nanoparticle in the bloodstream is characteristic of its surface. According to the literature, surface charge and attached targeting moieties are the key factors in the targeting process [2,3]. Nevertheless, nanoparticle absorption is a process not fully known and is often based on assumptions. According to the literature, nanoparticles are absorbed by endocytosis [2,4]. Receptor-associated particles are encapsulated in the cell membrane as they block the transporting protein forming an endosome. For effective targeting of nanoparticles against selected cell types, it is necessary to attach specific proteins that interact with receptors present only on the surface of malignant cells [3]. Such an approach requires a thorough knowledge of the structure of the protein and its active site and involves developing a method of attaching proteins to the surface of nanoparticles without compromising their form, which can often be problematic. An additional difficulty of this approach is the variety of tumour cells, and the types of specific receptors present on their surfaces. In the stage of rapid growth, the tumour creates a chaotic structure cut through with blood vessels created along with cell divisions and a growing demand for nutrients, mainly simple sugars and folic acid. In 1924, the German chemist Otto Warburg hypothesised that cancer cells have an increased demand for glucose due to changes in their metabolic pathway. A possible shutdown of mitochondria induces a shift of the sugar metabolic pathway in tumour cells to the glycolysis process. Although this process produces fewer energetic molecules of ATP than aerobic oxidation, the glucose demand grows approximately 200 times compared with that in initial, healthy cells. The phenomenon observed by the researcher has been called the Warburg effect and has been widely discussed in the literature [5,6,7,8]. Many years later, experiments have confirmed significant overexpression of the receptors responsible for the transport of glucose (GLUT) into cells [6,7,8]. Thus, it is assumed that the presence of glucose moieties on the surface of nanoparticles can be a sufficient targeting factor for the control of cancer cells [9,10,11,12]. This has led to an increased interest in polysaccharide nanoparticles as potential anticancer drug carriers [13]. It has also been supported by many successful and promising attempts to treat cancer cells with drug-loaded saccharide nanoparticles [14,15]. However, the experimental proof of polysaccharide nanoparticles undergoing close interaction and connexion with glucose transporter proteins remains weak.

Surface plasmon resonance (SPR) is being applied in studies of many types of interactions between analytes present in examined liquid samples and ligands immobilised on the surface of metals that exhibit affinity to the analyte. The SPR technique is one of the most frequently used techniques in the studies of protein–drug and drug–receptor interactions [16,17]. In addition, GLUT receptors have been successfully studied by SPR already. This technique has been used to examine the affinity of protein drugs that block them [18]. In this case, covalent binding of the GLUT1 protein to the surface of the SPR chip was used. Studies of the interactions of carbohydrates with specific blood proteins have been conducted, and low detection limits of below 0.01 mg mL^−1^ have been reached [19]. Lately, the detection of simple sugars with the use of SPR has been achieved in a good concentration range of millimoles, for such small-molecular target [20]. Finally, there are publications in which researchers conducted SPR studies of polymer and polysaccharide nanoparticles and their affinity to blood proteins such as heparin [21].

In this study, we are attempting to shed light on the matter of polysaccharide nanoparticle transport into cells and see whether GLUT proteins have an active role in this process. Namely, whether GLUT proteins interact with polysaccharide nanoparticles as with simple saccharides. We focused on two glucose transfer proteins: GLUT1 and GLUT4. GLUT1 is found in almost every human tissue and is responsible for basal glucose uptake. It also plays a role in glucose transport across blood tissue barriers [22]. GLUT4 is located mostly in muscle, fat, and heart tissue and is responsible for insulin-regulated glucose transport [22]. Both these transporter proteins have been suggested as targeting sites for novel smart therapies. For example, a study showed the possibility of blocking the active sites of these proteins by HIV protease inhibitors as a promising novel diabetes treatment [23]. A recent study published by Jagdale presented the use of GLUT1 targeting by lipid nanoparticles for cancer treatment [24]. These differences in location and function determined why these two specific proteins were selected for the investigation. Considering that there may be new anticancer drug delivery strategies based on targeting these transporter proteins, we aimed to determine whether these proteins interact with polysaccharide nanoparticles.

## 2. Results

The recorded sensograms of interactions between glucose transport proteins and simple saccharides and with dextran nanoparticles were prepared using the MP-SPR Navi Data Viewer and Ridgeview Instruments AB Trace Drawer 1.8 software and are shown in Figure 1, Figure 2, Figure 3, Figure 4, Figure 5 and Figure 6.

## 3. Discussion

Initial experiments on glucose and sucrose interactions confirmed that both saccharides exhibit affinity to both GLUT proteins. It was very difficult to obtain measurable results in the case of these saccharides because they are both very small molecules. Even though the equipment used was up to date, in most of the studied cases we needed to use a saccharide solution concentration of 250 mM to see a significant response. The exception was the interaction of glucose with the GLUT1 protein, which was observable at 4 mM. Nevertheless, at such high concentrations, the measured signal values were quite high, with maximums of up to 2 mdeg. The interactions were also measured to be very quick, with the peaks measured having only about 10 s duration. This may mean that the conditions of the experiment were suboptimal in terms of proper binding conditions of saccharides because the dissociation of the GLUT–saccharide complexes appeared to be very fast. The immobilisation of the GLUT proteins in more nature-imitating conditions, e.g., with other membrane proteins, parts of cell membranes, or even immobilisation of whole living cells could have given different results. In the case of experiments with nanoparticles, for both investigated proteins, similar concentrations gave a significant SPR response. The interaction resulted in much smaller signal peak values and longer interaction durations. The detachment of nanoparticles took much more time, which was mainly observed in the sensor regeneration part of experiments, which took approximately 10 min. This can be explained either by binding of a single nanoparticle to multiple protein particles or, rather unlikely, by different sites of proteins engaged in the interaction. The measured signals for glucose and sucrose resulted in affinities far from those already published, where reported Kd values are expressed in the millimolar range. Lundqvist and Lundahl experiments on GLUT–glucose affinity dependence on temperature resulted in measured Kd values of 56 mM at 5 °C to 26 mM at 42 °C [25]. Our affinity measurements (performed at room temperature) resulted in Kd values in the range of tens of moles, which is approximately thousand times higher than the values reported in the literature. This result comes from the nature of the SPR method itself, as much of this experiment, in which a small molecular target was bound by a much larger ligand. The SPR technique usually struggles in such experiments, and for precise calculations of Kd values different experimental designs should be applied, such as conjugation of the target molecule with magnetic or gold nanoparticles or use of secondary reporter proteins [26]. Such calculations were not, however, the purpose of this study. The maximum response for the 200 mM concentration of nanoparticles was recorded to be approximately 22 RU in the case of GLUT4, and 55 RU for GLUT1. These are small but significant signal values that show that interaction between glucose transporters and polysaccharide nanoparticles occurs. This means, that of the two examined proteins, GLUT1 exhibits a higher affinity to polysaccharide nanoparticles. It can be considered when planning, e.g., the dosing of drug-loaded nanoparticles depending on cancer type. The experiments for both GLUT proteins confirmed the participation of GLUT proteins in the process of nanoparticle internalisation by cells, as shown, e.g., by Wasiak et al. [10].

## 4. Materials and Methods

### 4.1. Materials

GLUT1 and GLUT4 whole active proteins were purchased from Abcam (Cambridge, UK). Saccharide analytical standards were purchased from Merck (Darmstadt, Germany). Dextran nanoparticles (average molecular mass: 40,000 Da) were prepared using a method described previously by our group [10]. In brief, 40 kDa dextran is oxidised by sodium periodate, which does not break the polymer backbone but opens some of dextran’s glucose rings, 20%. As a result, two aldehyde groups are formed in each of the oxidised glucose rings. Then, dodecylamine is attached to the aldehyde groups forming imine bonds, which are later reduced to amines by sodium cyanoborohydrate. These lipophilic side groups force Dextran chains to form nanoparticles. Their structure is not solid but rather cloud-like, the water content of such nanoparticles is about 90%. According to our research, approximately 10–11 dextran chains are employed to form a single nanoparticle [10]. In this paper, when we refer to the molar concentration of nanoparticles, we refer to the single 40 kDa dextran chain, not to the entire nanoparticle. After the process of synthesis, nanoparticles were measured in terms of size distribution using a Malvern Zetasizer Nano (Malvern Panalytical, Malvern, UK) and lyophilised. Reagents for protein immobilisation were purchased from Merck, Germany (salts for preparation of buffers, ethylenediamine hydrochloride), XanTec, Düsseldorf, Germany (EDC), and Thermo Scientific, Waltham, MA, USA (sulfo-NHS). SPR slides for the BioNavis MP-SPR apparatus were purchased from either BioNavis, Tampere, Finland or XanTec. For the dialysis of proteins and peptides, Slide-A-Lyzer MINI Dialysis units were purchased from Thermo Scientific, USA.

### 4.2. Equipment

Affinity experiments were performed using a fully automatic Multi Parameter Surface Plasmon Resonance apparatus (BioNavis NAALI, BioNavis, Finland) equipped with an autosampler for precise solution dosing. The processing of the recorded results was performed using MP-SPR Navi Data Viewer software, version 6.4.0.5. The visual representation of the sensograms was prepared using Ridgeview Instruments AB Trace Drawer 1.8 software.

### 4.3. Methods

#### 4.3.1. Immobilization of GLUT Proteins

Gold sensing chips grafted with a high-capacity hydrogel surface (in case of both proteins GLUT1 and GLUT4) were chemically modified by protein attachment. The chips were initially prepared with a condition buffer (100 mM NaOH + 2 M NaCl), and then carboxylic groups were activated with activation buffer (50 mM MES/NaOH, pH 5.0). Subsequently, the solution of protein in immobilisation buffer (5 mM MES/NaOH pH 5.0) was passed over the sensor surface for 20 min, followed by deactivation by quenching buffer (1 M ethanolamine/HCl pH 8.5). The sensogram representing the immobilisation process of GLUT1 performed using BioNavis NAALI equipment is shown in Figure 7. The prepared chips were then used in affinity experiments.

#### 4.3.2. Glucose and Sucrose Interaction Experiments

Solutions of saccharides (glucose and sucrose) in concentrations ranging from 4 mM to 1 M in the running buffer (PBS pH 7.4 + 0.05% Tween 20) were prepared, injected using a built-in autosampler, and passed over the surfaces of the sensing chips in the BioNavis MP-SPR apparatus. After each sample, regeneration buffer (10 mM glycine/HCl pH 2.4) was passed to detach saccharides bound to the protein on the surface, followed by a short injection of running buffer to stabilise the experimental conditions before the injection of the next saccharide solution. Every sample was examined in two repetitions where the sample concentrations were passed in an ascending and descending, manner. The results collected in the latter, were considered. This, as well as the use of a low-pH regeneration buffer, was conducted to ensure that the measured affinities were only of specific matter, instead of unspecific interactions. The general idea of the experimental design is presented in Figure 8.

#### 4.3.3. Polysaccharide Nanoparticle Experiments

Samples of nanoparticles were prepared in concentrations ranging from 0.5 µM to 200 mM by diluting the initial 200 mM solution by a factor of 0.2. The procedure of the experiments was the same as in the case of glucose and sucrose detection.

## 5. Conclusions

The Warburg effect has been known for 100 years, and his postulate gave us hope and helped in finding new targets in cancer metabolism and targets on cancerous cell membranes. However, experimental proof of polysaccharide nanoparticles undergoing close interaction and connection with glucose transporter proteins was weak, and the presented paper provides additional data to prove it. The obtained results show that the investigated glucose-transporting proteins exhibit affinity for dextran nanoparticles. This confirms the assumption that these proteins play an active part in polysaccharide nanoparticle endocytosis. GLUT1 is more sensitive to polysaccharide nanoparticles than GLUT4. This suggests that of these two proteins, GLUT1 is a preferable glucose transporter to be targeted in therapy using polysaccharide nanoparticle carriers. This, of course, depends on the type of cancer and the tissue the therapy is going to concern. The present study focused only on pure proteins and their activity while disregarding cell metabolism, the number of particular protein transporters in the cell membrane, and other crucial factors that should be considered when designing novel anticancer therapies. However, the results presented can be useful in the future in the design of new drug transport strategies.

## Figures and Tables

**Figure 1 ijms-24-16079-f001:**
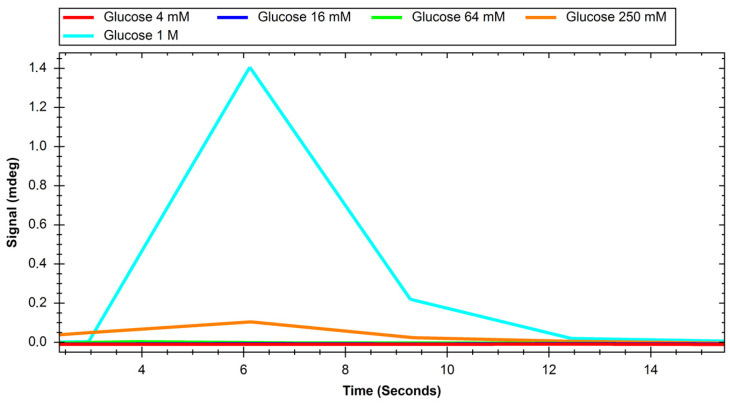
Response of GLUT1 protein to solutions of different concentrations of glucose.

**Figure 2 ijms-24-16079-f002:**
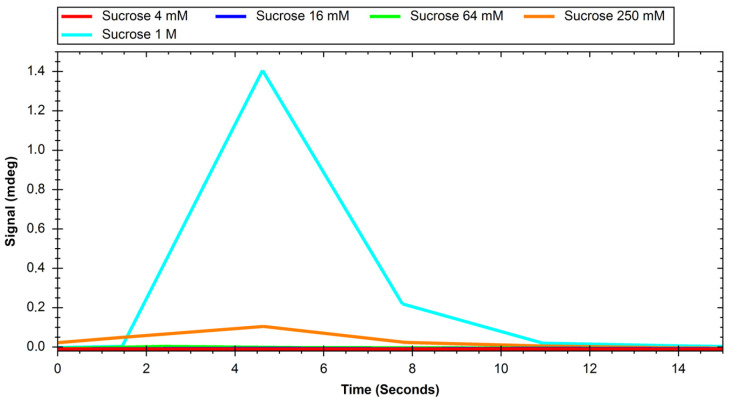
Response of GLUT1 protein to solutions of different concentrations of sucrose.

**Figure 3 ijms-24-16079-f003:**
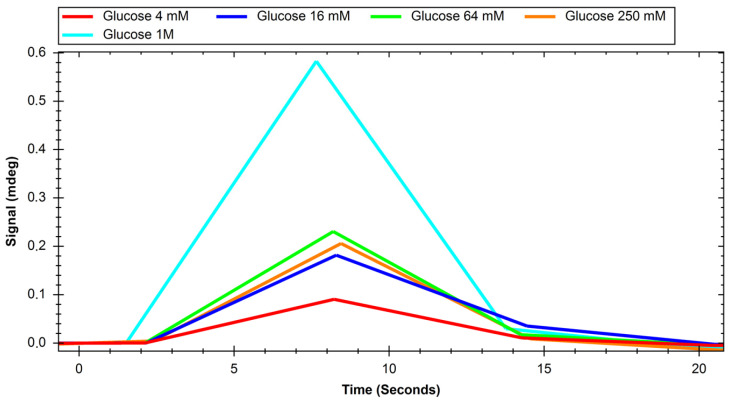
Response of GLUT4 protein to solutions of different concentrations of glucose.

**Figure 4 ijms-24-16079-f004:**
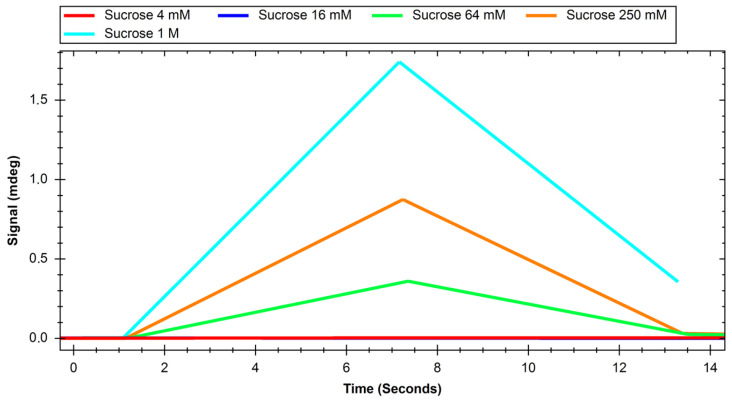
Response of GLUT4 protein to solutions of different concentrations of sucrose.

**Figure 5 ijms-24-16079-f005:**
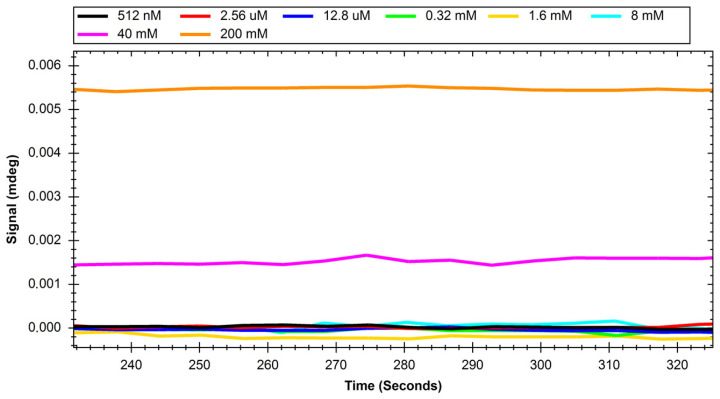
Response of GLUT1 protein to solutions with different concentrations of polysaccharide nanoparticles.

**Figure 6 ijms-24-16079-f006:**
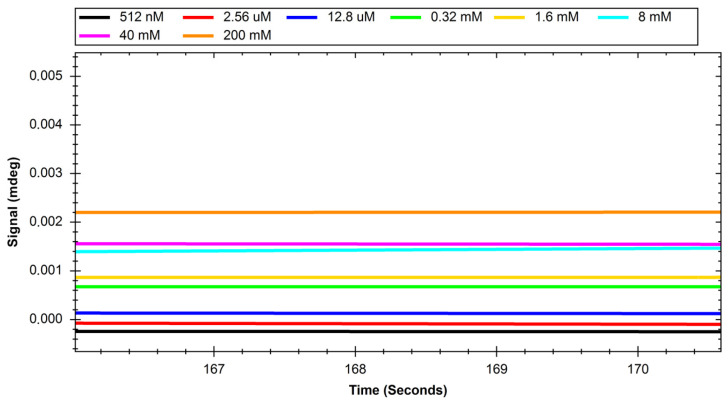
Response of GLUT4 protein to solutions with different concentrations of polysaccharide nanoparticles.

**Figure 7 ijms-24-16079-f007:**
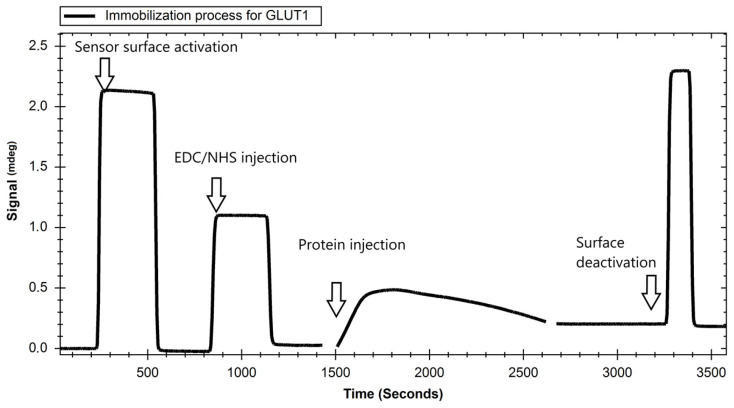
Immobilisation curve for the GLUT1 protein (BioNavis chip). After the process, the measured value increased by approximately 0.2 mdeg, which can be interpreted according to the sensor and apparatus specifications as 2000 RIU (response units). Furthermore, it can be understood that the result of the process was the immobilisation of approximately 2 ng of protein/mm^2^ of the sensor surface.

**Figure 8 ijms-24-16079-f008:**
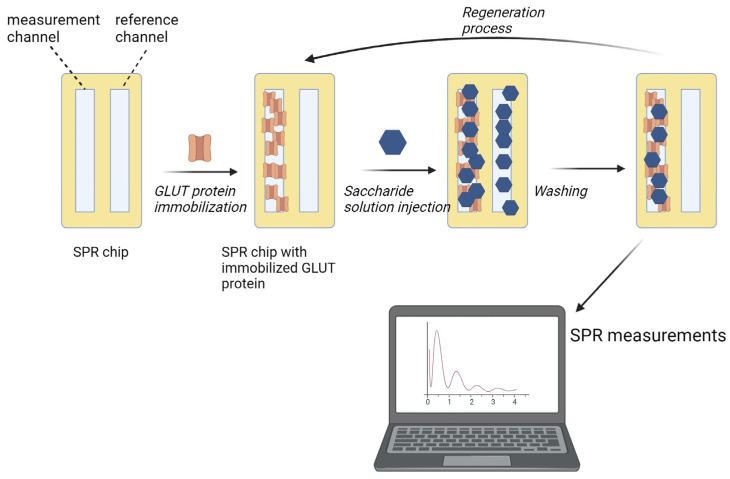
Scheme of the exemplary SPR experiment performed in this study.

## Data Availability

Data are contained within the article.

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
