# Peer review of "Study on Saccharide–Glucose Receptor Interactions with the Use of Surface Plasmon Resonance"

_ijms, 2023, doi:10.3390/ijms242216079_

Round 1
Reviewer 1 Report
Comments and Suggestions for Authors
The manuscript entitled "Study on saccharide – glucose receptor interactions with the use of surface plasmon resonance" presents fundamental studies that may be of great interest to the broader audience. However, the presented content is more suitable for a short communication than a full-length article. Some ambiguities should also be addressed to improve the quality of the paper.
Sucrose and glucose were used as reference compounds, but the measurements using them were carried out with a different instrument of lower sensitivity. It seems more appropriate to present data obtained with the same instrument for comparison purposes.
The association, dissociation, and equilibrium (KD) constants should be derived from sensograms to better understand the interaction between the saccharide particles and chosen glucose receptors.
A 'condition buffer' was used to prepare the SPR chip. However, the solution of 100 mM NaOH and NaCl does not meet the definition of a buffer solution.
Minor editorial and linguistic corrections are required, so careful revision is advised. For example, the main text of the results section is in bold; the upper scripts are missing; the expression on page 2, line 63 should be mitochondrial shutdown/shutdown of mitochondria, etc.
Comments on the Quality of English LanguageThe overall English quality is good; only minor corrections are required, as already stated.
Author Response
Dear Reviewer
Thank you for constructive remarks. Below are my answers.
ad 1. I agree, the measurements for the glucose and sucrose have been repeated on the BioNavis apparatus and results attached.
ad 2. Since the overall sensitivity obtained by the system is quite low, the equilibrium constant measured by it is very unprobable. I mentioned it in the discussion section.
ad 3. It is truly not a buffer in its standard meaning, however technical documentation of the equipment as well as some literature calls it so. The unfortunate term has been replaced with "condition solution".
ad 4. Minor editing corrections have been applied.
Reviewer 2 Report
Comments and Suggestions for Authors
The figures in the manuscript should be prepared much more professionally to improve the soundness of the manuscript. And more discussions should be given to each figure.
Comments on the Quality of English LanguageThe English Language of this manuscript is not bad.
Author Response
Dear Reviewer
Thank you for the remark. Most of the figures have been changed, one additional figure has been added to visually describe the experiment design.
Reviewer 3 Report
Comments and Suggestions for Authors
Following a thorough review of the manuscript entitled "Study on saccharide – glucose receptor interactions with the use of surface plasmon resonance." by Maciej Trzaskowski et al., submitted to the International Journal of Molecular Sciences, it is evident that the manuscript's thematic content aligns adequately with the journal's scope and addresses pertinent issues within current research. The manuscript can be accepted for publication after the authors address the following issues:
1. Abstract: Authors may add quantitative or qualitative results of experiments to the Abstract to present the highlights and scientific results of the study.
2. The article does not clearly explain its technical methods and principles. It is recommended to insert a scheme to represent the modification process and reaction mechanism of the SPR sensing chip surface, as this is the core topic of this manuscript.
3. The authors should provide a detailed comparison of their work's performance with existing literature, specifically addressing parameters such as detection limit, sensitivity, specificity, accuracy, response time, and cost-effectiveness?
4. In Figure 1-5. Signal responses at low concentrations appear to overlap or be lower than baseline. How does the author determine the authenticity of the signal, and what is the signal-to-noise ratio (S/N) when doing quantitative measurements? In Figure 1-5. Signal responses at low concentrations appear to overlap or be lower than baseline. How does the author determine the authenticity of the signal, and what is the signal-to-noise ratio (S/N) when doing quantitative measurements?
5. SPR measures changes in the refractive index caused by molecular interactions on metal surfaces via a surface plasmon waves. The measurement results in this paper, the measurement values must be presented strictly. Normally, in science, the measurement is conducted at least 3 times so that you can have an average value and plus/minus the standard deviation. What is the standard deviation of this measurement? It would be nice to include that in the text.
6. Figure 7. Part of the immobilisation curve for the GLUT1 protein. The authors should indicate the protein immobilization process in the figure. Please reshape it.
7. It is well known that SPR sensors have superior sensing capabilities. However, when a low pH buffer is used in the desorption experiment during the regeneration process, will the carboxylated polysaccharide on the surface of the sensing chip be destroyed during this process? How many times can it be regenerated?
8. The authors should highlight some potential future applications in the conclusion.
Author Response
Dear Reviewer
Thank you for constructive remarks. Below are my answers.
ad. 1. Abstract has been improved, some quantitative results have been added.
ad 2. A scheme of the experiment design has been added.
ad 3. There is some comparison in the discussion section. Generally, our system exhibited quite low sensitivity so I could only say that measured results can be rather qualitative. Much better detection limits have been presented elsewhere in case of simple sugar detection, however no experiments showing both simple sugars and nanoparticles on SPR have been, to my knowledge, presented yet.
ad 4. The measurements mentioned were in fact at the noise level, though I concluded only two or three used concentrations gave significant responses.
ad 5. As I described in the methods section, each sample series has been measured two times: in ascending and descending order in terms of concentration. The results taken into consideration were taken from the latter measurement series. I do not present results as numerical points, but rather as signal strength curves. This is why I did not calculate the standard deviations.
ad 6. The figure has been replaced with a curve representing the whole immobilization process
ad 7. This conditions for regeneration have been selected before the experiments and were found to be the most effective in detachment of sugar molecules from the protein while still not damaging the protein itself. It seems that many regeneration processes can be conducted with this solution since the baseline of the experiment was stable.
ad 8. Some conclusions have been added.
Reviewer 4 Report
Comments and Suggestions for Authors
Manuscript review No: ijms-2633025
Title: Study on saccharide – glucose receptor interactions with the use of surface plasmon resonance.
Authors: Maciej Trzaskowski, Marcin Drozd, and Tomasz Ciach
A. Overview
1. In this manuscript the authors report on a study on the matter of polysaccharide nanoparticle transport into the cells and examin whether GLUT proteins have an active role in this process.
2. The contents are expressed clearly; the manuscript is well organized, and it is written in reasonable English.
Though, reading of the manuscript is required as several misprints
3. Many references are more than 5 years old. A few references on the use of surface plasmon resonance can be found in the literature, such as
On the Use of Surface Plasmon Resonance Biosensing to Understand IgG-FcγR Interactions, 2021, doi: 10.3390/ijms22126616
The revelation of glucose adsorption mechanisms on hierarchical metal–organic frameworks using a surface plasmon resonance sensor. 2023, https://pubs.rsc.org/en/content/articlelanding/2023/tb/d3tb00138e/unauth
and references therein.
4. As long as my knowledge, the work presented is original. However, is not a remarkable new issue.
B. Detailed analysis.
Abstract: must be clear and objective. State briefly what you did, how did you do it, the quantitative results you and state clearly the novelty of your work.
1. INTRODUCTION: provides an interesting approach to the subject and there are up to date references.
C. Overall assessment
The work presented here is very interesting and has potential for further developments.
In my opinion the manuscript can be accepted for publication after revision.
D. Review Criteria
1. Scope of Journal
Rating: Medium
2. Novelty and Impact
Rating: Medium
3. Technical Content
Rating: Medium
4. Presentation Quality
Rating: Medium
Comments on the Quality of English Language
The contents are expressed clearly; the manuscript is well organized, and it is written in reasonable English.
Author Response
Dear Reviewer
Thank you for the remarks. I expanded the cited literature section. I have added the publications suggested by you as well as some other papers.
Round 2
Reviewer 1 Report
Comments and Suggestions for Authors
The response to the review is comprehensive, and the changes introduced within the manuscript are sufficient.
Author Response
Thank you for the positive review.
Reviewer 2 Report
Comments and Suggestions for Authors
The authors should give more discussions based on Figures 2-6. I think the most important part of a paper is the discussion part. However the authors only give a very short discussion to the results.
Comments on the Quality of English LanguageOverall the quality of English is acceptable. The authors should check the manuscript carefully. For example, the last sentence of introduction, there should be no comma after Considering.
Author Response
Thank you for the review. The Discussion section has been significantly expanded. The whole article has been carefully checked and corrected in terms of language errors.
Reviewer 4 Report
Comments and Suggestions for Authors
The contents are expressed clearly; the manuscript is well organized, and it is written in reasonable English.
Comments on the Quality of English LanguageManuscript review No: ijms-2633025 - REV1
Title: Study on saccharide – glucose receptor interactions with the use of surface plasmon resonance.
Authors: Maciej Trzaskowski, Marcin Drozd, and Tomasz Ciach
A. Overview
1. In this manuscript the authors report on a study on the matter of polysaccharide nanoparticle transport into the cells and examin whether GLUT proteins have an active role in this process.
2. The contents are expressed clearly; the manuscript is well organized, and it is written in reasonable English.
3. As long as my knowledge, the work presented is original. However, is not a remarkable new issue.
B. Overall assessment
- The work presented here is very interesting and has potential for further development.
- The authors answered all questions and queries and modified the manuscript according to the reviewers’ comments.
- In my opinion the manuscript can be published
C. Review Criteria
1. Scope of Journal
Rating: Medium
2. Novelty and Impact
Rating: Medium
3. Technical Content
Rating: Medium
4. Presentation Quality
Rating: Medium
Author Response
Thank you for the positive review. Minor English mistakes have been corrected.